# Catalytic Systems in the Reduction of Nitrogen Oxide Emissions in Diesel-Powered Trucks

Jessimon Ferreira [1], Dana I. Andrade [2], Maria E. K. Fuziki [3], Lariana N. B. de Almeida [3], Leda M. S. Colpini [4], Giane G. Lenzi [2] and Angelo M. Tusset [1,*]

1   Department of Production Engineering, Federal University of Technology-Paraná, Paraná-Rua Doutor Washington Subtil Chueire St. 330, Ponta Grossa 84017-220, Brazil; jessimon@alunos.utfpr.edu.br
2   Department of Chemical Engineering, Federal University of Technology-Paraná, Paraná-Rua Doutor Washington Subtil Chueire St. 330, Ponta Grossa 84017-220, Brazil; dana@alunos.utfpr.edu.br (D.I.A.); gianeg@utfpr.edu.br (G.G.L.)
3   Department of Chemical Engineering, State University of Maringá, Colombo Ave. 5790, Maringá 87020-900, Brazil; mariafuziki@alunos.utfpr.edu.br (M.E.K.F.); beraldolariana@gmail.com (L.N.B.d.A.)
4   Department of Chemical Engineering, Federal University of Paraná, Rua Doutor João Maximiano 426, Jandaia do Sul 86900-000, Brazil; ledasaracol@ufpr.br
*   Correspondence: tusset@utfpr.edu.br

**Abstract:** In recent years, the number of motor vehicles in circulation has increased in proportion to Brazil's economic growth, resulting in an increase in emissions of toxic gases from combustion, such as nitrogen oxide, particulate matter, carbon dioxide and volatile organic compounds, among other polluting compounds. This type of pollution has its impacts potentiated in large cities, accumulating due to the configuration of streets and buildings in large urban centers, and can even penetrate indoor environments, having harmful effects on the health of residents. To minimize the emission of these gases, catalytic converters can be used in the vehicle exhausts. Catalytic converters are a promising technology used to reduce exhaust emissions from the engine. In this context, this paper presents an overview of the emission of toxic gases by heavy transport powered by diesel oil and the influence of the use of automotive catalysts in reducing the emission of toxic gases. Additionally, a proposal for monitoring the useful life of automotive catalysts is presented through an electronic sensing system, which makes it possible to determine the catalyst efficiency and the appropriate point for its reactivation or replacement.

**Keywords:** automotive catalysts; emission of toxic gases; temperature sensor





## 1. Introduction

In recent years, the number of motor vehicles in circulation has increased, and consequently, there has been an increase in emissions of gases from combustion. Thus, it is necessary to monitor the emissions of these gases and study methods and technologies to reduce such emissions [1–3].

Road transport is a significant source of urban pollutant emissions and will likely continue for decades [4]. Diesel engines have the advantages of high engine efficiency and low fuel consumption, contributing to their wide use. However, the exhaust gas from the combustion of these engines contains a large amount of nitrogen oxide ($NO_x$), particulate matter (PM), carbon dioxide ($CO_2$) and volatile organic compounds (VOC), among other polluting compounds [5,6].

The transport sector ranks second in the production of global $CO_2$ emissions, with a range of 22% [7,8]. According to Giechaskiel et al. [9], data from 2008 show that although vehicles with diesel engines represent less than 5% of the vehicle population in California (USA), China, and Brazil, these vehicles contributed from 44% to 57% of all pollutants emissions, such as particulate matter from road traffic and $NO_x$ gas emissions. In addition to $NO_x$, emission from motor vehicles contains greenhouse gases.

In this way, pollution in large cities has become a matter of urgency. This concern is further compounded by the fact that in urban centers, due to their configuration, it is common to find areas of street canyons (streets surrounded by buildings on both sides), in which ventilation is reduced, favoring the accumulation of pollutants and particles which may be very harmful to human health [10,11]. These pollutants can enter buildings through windows and doors or even through cracks in them, thus reducing indoor air quality [10,12]. Lawrence et al. [13], for example, identified a positive correlation between the outdoor and indoor concentration of NO and $NO_2$ in an urban site in India, and correlated this to the traffic pollution [13]. In this sense, Salonen et al. [14], through their review study, gathered reports that indicated that schools located close to highways, industrial areas, or in urban centers had considerably higher concentrations of $NO_2$ than those located in less intense traffic locations, suggesting the level of pollution outdoors can affect indoor air quality [14]. This scenario stimulated the search for ways to mitigate atmospheric pollution, some of which are quite innovative, such as the incorporation of photocatalysts in building materials in order to promote the reduction in $NO_x$ present in the air [15,16]. Even so, in view of the large contribution of heavy-duty vehicles (HDVs) for $NO_x$ pollution, it is estimated that the application of Euro VI standards could lead to a substantial reduction in $NO_x$ emissions (up to 80–90% in comparison to 2040 baseline) in places where these standards have not yet been fully implemented [17]. The addition of pollutant-conversion technologies by the action of catalysts (such as Selective Catalytic Reduction (SCR), Ammonia Slip Catalyst (ASC) and Diesel Oxidation Catalysts (DOC)) to heavy duty trucks exhaust has promising potential to contribute to meeting these emission limits [18].

Catalytic converters are a technology used to reduce gas emissions from the engine. A catalytic converter (or simply catalytic converter) is a device that converts toxic gases from combustion into other non-harmful gases. Catalysts are highly active and efficient in reducing emissions, and for this reason, they were developed for use in different types of vehicles. Today, many car manufacturers can meet emission standards through the use of catalytic converters. The lifespan of catalytic converters is up to 36 months, so they must be replaced at least every three years [19,20].

SCR is the most promising technology in reducing $NO_x$ emissions from combustion in boilers and diesel engines [21,22]. The SCR system uses ammonia as a reducing agent. An aqueous solution of urea is injected onto the catalyst's surface and, when it comes into contact with the hot exhaust gas, it evaporates and decomposes into ammonia $NO_3$, which promotes the reduction of $NO_x$ into nitrogen $N_2$ and water $H_2O$. The conversion of $NO_x$ in an SCR system occurs at a high rate and is highly dependent on the catalyst temperature [23–25].

According to Mardani et al. [3], understanding the relationship between $CO_2$ emissions and economic growth can help to develop energy policies and energy resources sustainably. In the study developed by Vieira et al. [15], the authors evaluated the influence of extreme events on electricity consumption and Gross Domestic Product (GDP). Figure 1 shows data from 1970 to 2014 for GDP per capita, per capita $CO_2$ emissions, and electricity consumption in Brazil.

The authors identified that, over the years, there was an increase in all the variables studied, which indicates that there is a correlation between them. The authors also observed that, between 2011 and 2014, there was a drop in the values of $CO_2$ emissions and GDP. This study emphasizes the importance of monitoring and studying the gases present in pollutant emissions, as they can affect the health of living beings, the ecosystem, and the economy.

In this context, this study presents the influence of automotive catalysts in reducing the emission of toxic gases from the internal combustion of diesel engines. The study also provides a proposal for monitoring the useful life of automotive catalysts through sensing, which makes it possible to determine the efficiency of the catalyst, as well as the appropriate point for its reactivation or replacement.

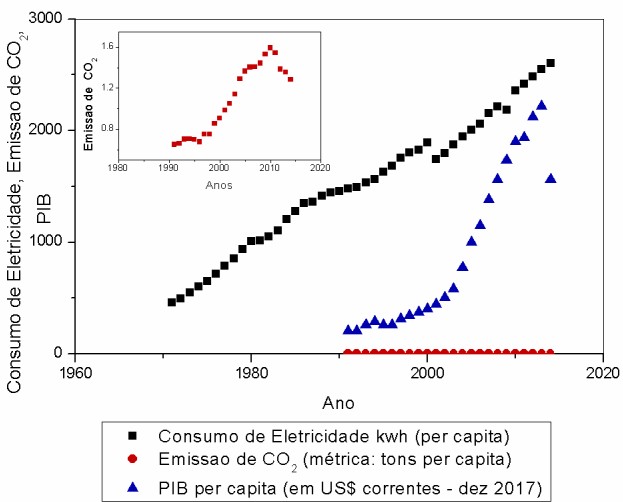

**Figure 1.** Gross Domestic Product (GDP), CO₂ emission, and electricity consumption in Brazil in per capita values in a time series [26].

## 2. Pollutant Emission by Diesel Vehicles in Brazil

Since 2012, Brazil has followed the EURO V standard, which provides information for implementing SCR systems for heavy vehicles that use diesel as fuel. The adoption of this technology has contributed to reducing pollutant gas emissions over the years. However, there are still many vehicles with obsolete technology in circulation. In addition, the increase in the vehicle fleet in Brazil, mainly in metropolitan regions, caused a rise in the number and extent of traffic jams and, consequently, in the release of pollutants into the atmosphere [27,28].

The creation of PROCONVE (Program for Control of Air Pollution by Motor Vehicles) by CONAMA (National Council for the Environment) in 1986 allowed Brazil to make progress in controlling emissions from motor vehicles. After the first phase of the program, in 1992, a reduction of approximately 70% was achieved for pollutants emitted by cars. The main objective of PROCONVE is to reduce pollutant emissions from new vehicles through the establishment of emission limits. Thus, with the progressive implementation of phases, technological improvements are induced in the automotive industry to reduce emissions from new vehicles placed on the market [29,30].

According to Benvenutti et al. [31], within the energy sector's greenhouse gas emissions, road transport is responsible for 91% of total CO₂ emissions. In Figure 2, the variation in CO₂ emission and the variation in the consumption of petroleum-derived diesel oil are presented, considering only vehicular activity.

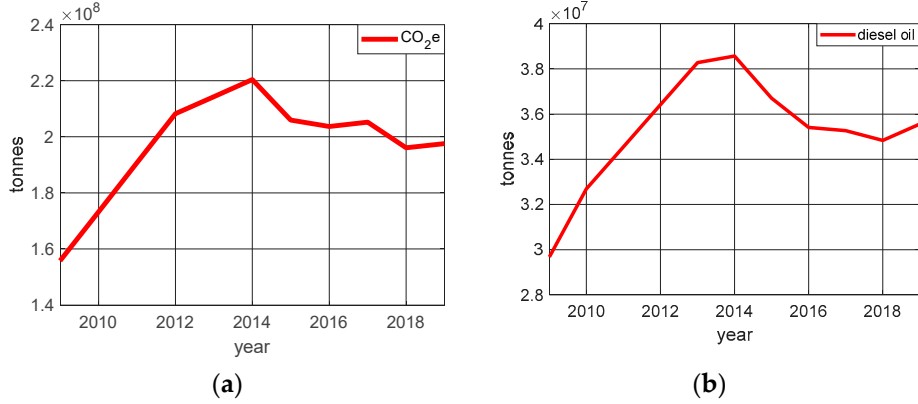

(**a**)　　　　　　　　　　　　　　　　　　　　　　(**b**)

**Figure 2.** Relation of the inclusion of catalysts in vehicle exhausts: (**a**) CO₂ emission (**b**) diesel oil consumption.

We can see in Figure 2 that the introduction of catalysts in vehicle exhausts from 2012 onwards, combined with the reduction in diesel consumption, contributed to a reduction in the growth rate of $CO_2$ emissions.

Figure 3 shows the number of truck-type vehicles in circulation in Brazil, as well as their rate of degradation according to the time of use.

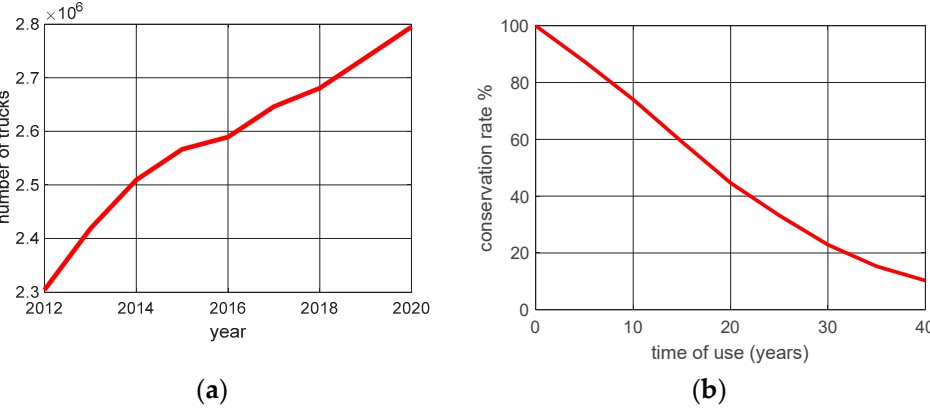

**Figure 3.** The number of trucks emitting toxic gases and their state of conservation. (**a**) The number of trucks. (**b**) State of conservation.

By analyzing the data in Figure 3a, we can observe a growth in the truck fleet in Brazil between 2012 and 2020. Considering the growth as linear, we can estimate a truck fleet of approximately 2,837,452 in 2022. According to CONAMA resolution 490/2018 [32], from 2023, new models of trucks and other vehicles will have new maximum limits for exhaust gas emissions, in addition to the mandatory installation of on-board diagnostic systems and measurement of emissions in real traffic. However, after the resolution, 2,837,452 trucks will still be running with unmonitored catalytic converter exhausts. Control of catalytic converter replacement at the appropriate time will depend solely on the owner. The automotive catalytic converter should be replaced approximately every three years, which according to Figure 2b, represents a vehicle with approximately a 90% conservation status.

By analyzing the results presented in Figure 3a,b, we can see that there was a reduction of approximately 20% in $CO_2$ emissions per truck compared to the number of trucks in 2012 (before the catalyst was mandatory) and 2019 (use of the catalyst in large part of the fleet).

Considering the 2019 fleet, it can be estimated that a vehicle emitted approximately 90,374 tons of $CO_2$ per year. Making a projection of these data, considering a scenario where the maintenance or replacement of the catalyst is not performed, we can estimate a possible emission of more than 250 million tons of $CO_2$ in 2022 by the fleet that will be in circulation. According to an estimate by the World Health Organization (WHO), 4.2 million deaths occur every year as a result of exposure to polluting compounds present in the air, a fact that characterizes air pollution as the fourth largest risk factor for death in the world [33].

## 3. Sensing the Catalytic Activities in the Vehicle Exhaust

Monitoring of the catalyst's efficiency in reducing toxic gas emissions from the vehicle's exhaust can be carried out in two ways, namely by controlling the catalyst temperature, or by determining the amount of ammonia at the output of the catalyst.

### 3.1. Monitoring by the Temperature Sensor

The conversion of $NO_x$ into $N_2$ and $H_2O$ is highly dependent on the catalyst temperature; that is, it is susceptible to engine operating conditions [25]. For the temperature values to be corrected throughout the process, it is necessary to integrate sensing elements and electronic control devices. The most commonly used temperature sensors are the thermocouple and the RTD (Resistance Temperature Detector) sensors. Figure 4

shows the temperature variations in an exhaust gas treatment system of an internal combustion engine.

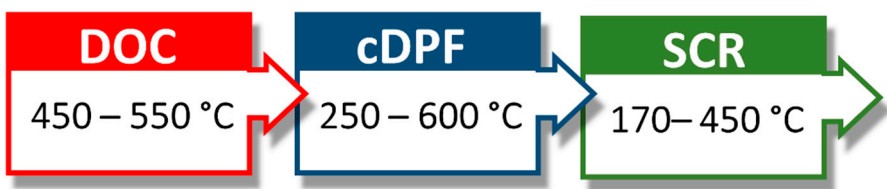

**Figure 4.** Temperature values necessary for the performance of the gas treatment system.

### 3.2. Monitoring by NO$_x$ Sensor

In SCR systems, ammonia (NH$_3$) is used as a reducing agent when converting NO$_x$ gases. In practice, NH$_3$ cannot be transported directly in vehicles, so an aqueous urea solution is injected into the system, which is decomposed into NH$_3$. However, if the urea concentration in the system is low, a reduction in NO$_x$ gases will be insufficient. On the other hand, if the urea concentration in the system is high, it causes NH$_3$ to escape, making this compound a secondary pollutant. To avoid this problem and control NO$_x$ emissions, the use of NO$_x$ sensors has become common in vehicles. The purpose of these sensors is, by means of the NO$_x$ dosage, to control the amount of NH$_3$ in the catalyst, and thus, proceed with the correct adjustment of the urea dosage in the stoichiometric proportion [34,35]. Figure 5 shows the graphic representation of the sensor.

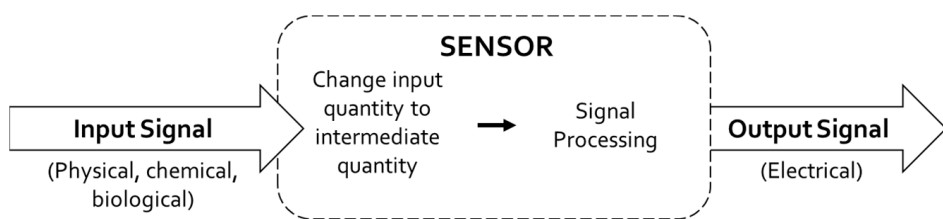

**Figure 5.** Graphic representation of the NO$_x$ sensor.

The sensor consists of two measuring Nernst cells. In addition to controlling the reactions within the chambers, the oxygen concentration in each cavity is also controlled by changing the pumping currents and comparing the reference cell potential with a setpoint value. To obtain the desired concentration, oxygen is pumped into or out of the cavities [36]. Figure 6 provides a schematic representation of the NO$_x$ electrical sensor.

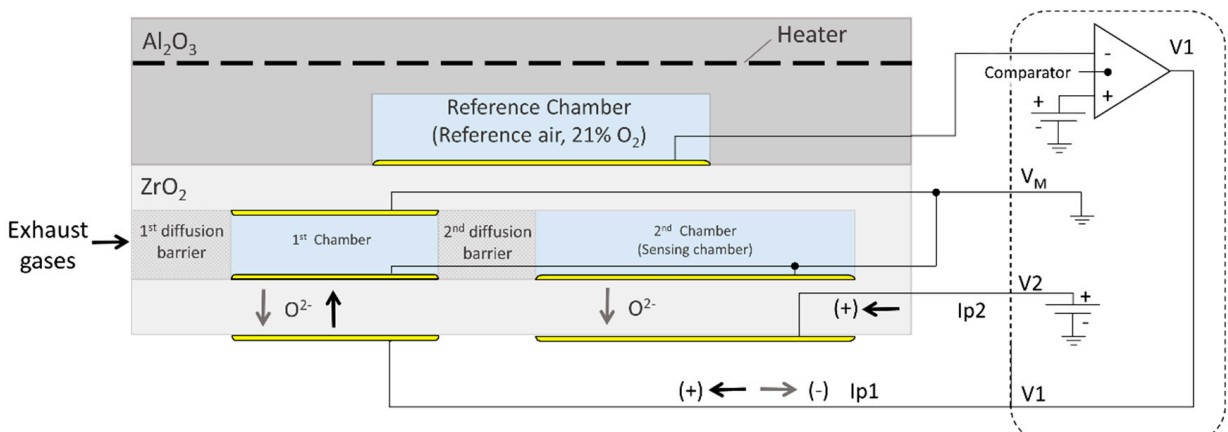

**Figure 6.** NO$_x$ sensor wiring diagram.

## 4. Performance Sensing Proposal of the SCR Catalyst Used in Vehicle Exhausts

For technical and legal reasons, modern exhaust after-treatment systems must be monitored [37]. The proposed mode of sensing comprises an electronic device composed of a microcontroller, $NO_x$ gas sensors, EGTS (Exhaust Gas Temperature Sensor) and an LCD display.

Through sensing, it will be possible to estimate the performance of the SCR catalyst and its useful life, in addition to diagnosing if there is a failure in the activation of the SCR catalyst. The data make it possible to anticipate maintenance solutions, thus preventing emissions from reaching a critical level of toxicity.

Another variable that was evaluated is the internal temperature of the SCR catalyst, which has a minimum difference of 10% between the inlet and outlet of the gases. The minimum temperature required for the process is in the range of 170 °C to 200 °C. Suppose the exhaust does not reach the appropriate temperature for the activation process of the SCR catalyst. In that case, the reduction of $NO_x$ gases will not occur due to the inefficiency of the chemical reaction between the noble metals of the catalyst, the gases from the exhaust, and the liquid urea injected at the inlet of the catalytic device. Failure in the activation process can be due to:

✓ SCR, DOC and cDPF (Catalyzed Diesel Particulate Filter) filter devices showing internal or external cracks;
✓ SCR devices, DOC and cDPF filter clogged by soot;
✓ Failure to control the air/fuel mixture;
✓ Low fuel-burning power in the combustion chamber;
✓ Pipe leaks;
✓ Low chemical reaction (which is exothermic) of DOC and SCR.

Figure 7 shows the sensing proposal used in the treatment system for exhaust gases from diesel engines.

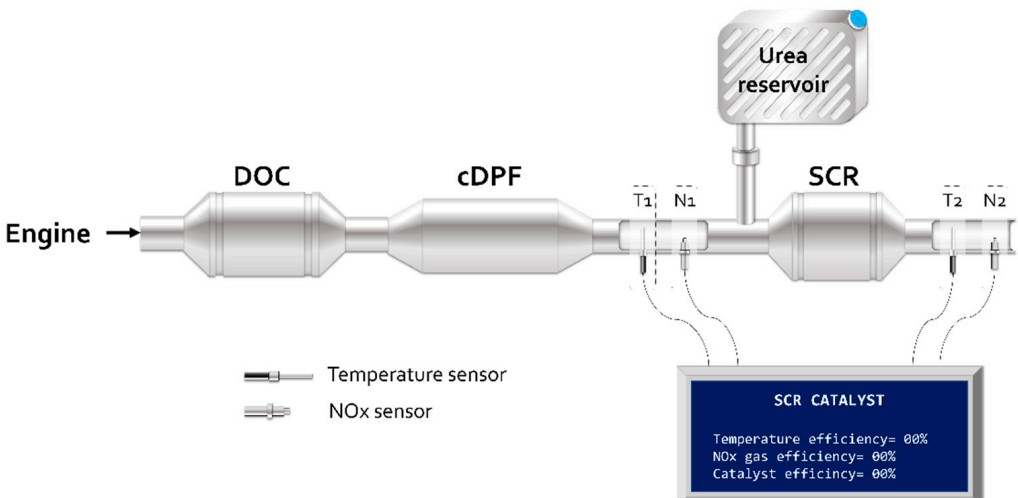

**Figure 7.** Implementation of the onboard monitoring system.

In Figure 7, the integration of $NO_x$ sensors, temperature, and the electronic device for conditioning and data acquisition in the post exhaust gas treatment system is presented. Figure 8 provides the electrical diagram of the temperature monitoring sensor and the $NO_x$ sensor, used to analyze the SCR catalyst performance.

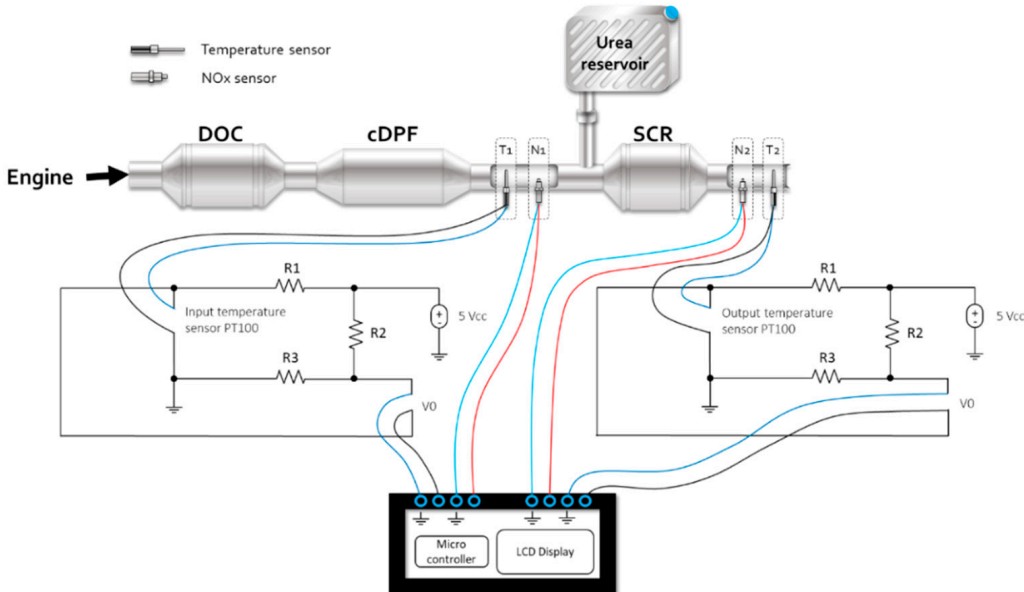

**Figure 8.** Integration of electronic components in the SCR catalyst monitoring system.

The $T_1$ and $T_2$ temperature sensors provide the electrical signals corresponding to the temperature values measured at the ends of the SCR catalyst. Figure 9 shows the catalyst failure analysis flowchart of temperature monitoring.

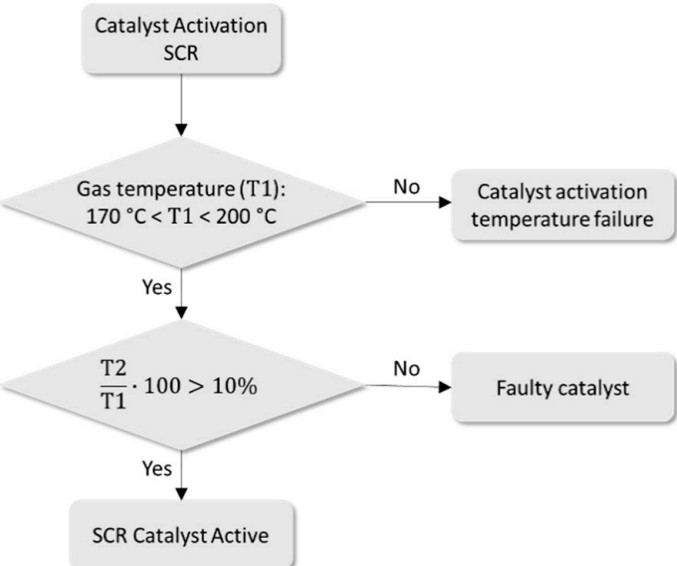

**Figure 9.** Flowchart for temperature monitoring.

Figure 10 presents the flowchart for evaluating catalyst failure by monitoring the reduction of $NO_x$ gases.

For $NO_x$ sensors, if $N_1 = N_2$ then the catalyst is faulty.

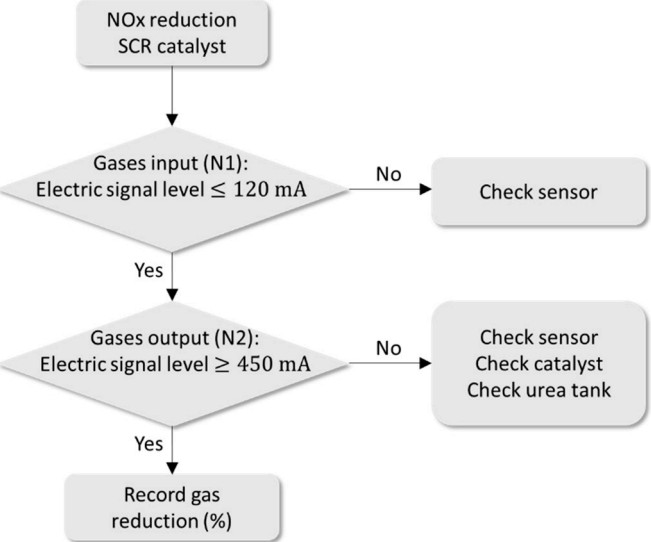

**Figure 10.** Flowchart for monitoring the NO$_x$ gas sensor.

*Computer Simulation of the Monitoring System*

To obtain the temperature a PT 100—platinum (Pt)-based sensor with 100 Ω resistance at 0 °C was used, as represented in Equation (1).

$$T = \frac{\frac{R_s}{R_0} - 1}{\emptyset} \tag{1}$$

where $R_2$ = Internal resistance of the sensor as a function of the variable $T$; $R_0$ = Value of the minimum range of the PT100 sensor; $T$ = Temperature of the internal catalyst gases; $\emptyset$ = Pt (platinum) sensor temperature coefficient 100 (0.00385 IEC 60751).

For the SCR catalyst to be considered active, a temperature differential of the gases between the inlet and outlet of the device is required. This differential is the result of the exothermic reaction inside the catalyst. For the catalytic converter to be active, this differential must be at least 10% greater during the exit of the gases.

The temperature differential can be obtained with Equation (2).

$$D_t = \left(\frac{T_s}{T_e} - 1\right) 100 \tag{2}$$

where $D_t$ represents the catalyst temperature differential, $T_e$ represents the temperature of the gases at the inlet of the catalyst, and $T_s$ the gas temperature in the catalyst exit.

Another method to determine the temperature of the catalyst gases is to identify the potential difference (ddp), where the output voltage ($V_{out}$) is collected at the resistive bridge. By identifying the variable R$_s$ (PT 100 sensor resistance), it is possible to identify the ddp at the output ($V_{out}$) of the bridge.

$R_1$, $R_2$, $R_0$ and $V_{in}$ values are considered constant in the resistive bridge, while $V_{out}$ can be obtained using Equation (3).

$$V_{out} = V_{in} \left( \frac{R_s}{R_1 + R_2} - \frac{R_0}{R_2 + R_0} \right) \tag{3}$$

The internal resistance of the sensor can be obtained from Equation (4), as a function of ddp $V_{out}$ of the resistive bridge.

$$R_s = R_1 \left( \frac{V_{out}(R_2 + R_0) + V_{in}R_0}{V_{in}(R_2 + R_0) - V_{out}(R_2 + R_0) - V_{in}R_0} \right) \tag{4}$$

Figure 11 shows a graph of the temperature variation ($T$) of a PT 100 sensor, considering Equations (1), (3) and (4), for $V_{in}$ = [0:5] Volts.

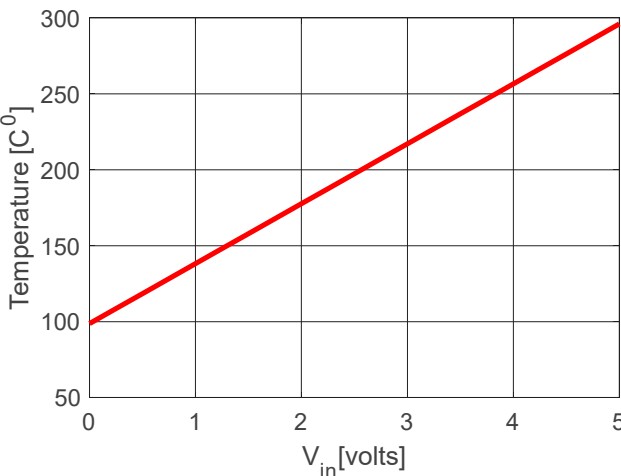

**Figure 11.** Temperature variation (*T*) of a PT 100 sensor versus input voltage.

To evaluate the efficiency of the SCR catalyst in the reduction of $NO_x$ gases, the electrical response of the sensors was considered, which were installed at the inlet and outlet of the gases in the catalyst, to perform an efficiency calculation in the reduction of gases. For an optimal state of reduced $NO_x$ gas emissions, the electrical output signal of the $NO_x$ sensor must have a stoichiometric value of 450 mV. For a low-efficiency reduction, the value indicated is 120 Mv, which is the value that the sensor can identify at the inlet of the catalyst immediately after combustion.

Equation (5) presents the calculation implemented to identify the value, as a percentage, of the reduction of $NO_x$ gases in relation to the inlet and outlet of gases in a diesel SCR catalyst.

$$E_\% = \left( \frac{ddp_{out} - ddp_{in}}{span} \right) 100 \tag{5}$$

where $ddp_{out}$ represents the electrical voltage value in mV presented by the output sensor of gases in the catalyst, $ddp_{in}$ represents the constant value of 120 Mv and *span* represents the theoretical range of the sensor for measuring $NO_x$ gases. For Equation (5), the value is a constant of 330.

Another possibility for analyzing the efficiency of the reduction of $NO_x$ gases in an SCR catalyst is to evaluate the ddp signal provided by the sensor installed at the output of the SCR catalyst, using the desired value of 450 mV as a reference, using Equation (6).

$$E_\% = \left( \frac{ddp_{out}}{ddp_w} \right) 100 \tag{6}$$

where $ddp_{out}$ represents the electrical voltage value in mV presented by the gas output sensor in the catalyst and $ddp_w$ represents the constant value of 450 mV.

In Figure 12, we can see the electronic circuit diagram of the efficiency of the SCR catalyst in relation to $NO_x$ gases.

Figure 13 shows the results for Equations (5) and (6), obtained in simulations in the Proteus virtual environment.

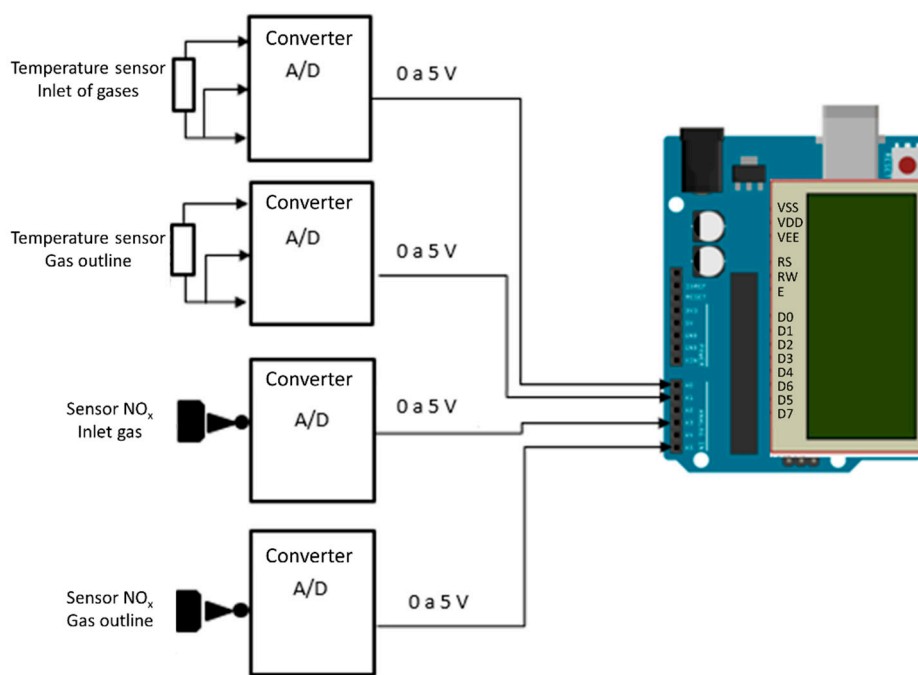

**Figure 12.** Electronic circuit for virtual simulation.

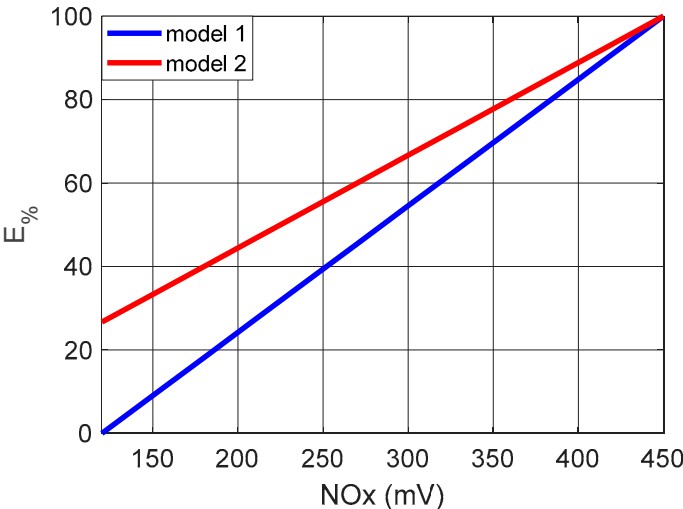

**Figure 13.** Analysis of catalyst performance for $NO_x$ variations.

As shown in Figure 13, the efficiencies calculated using Equations (5) and (6) present curves of equal evolution and can be used in parallel to guarantee robustness in the evaluation of the catalyst's performance.

## 5. Conclusions

The results presented demonstrate the relationship of the emission of toxic gases with exhaust systems without the catalytic system or with failure in the catalyst. The levels observed are worrying for the respiratory health of the population. As can be seen, with the proper use of the catalyst, it is possible to reduce the release levels of $NO_x$, PM, $CO_2$, and VOC, among other polluting compounds. Additionally, with the proposal of the catalyst efficiency sensor, this paper proposes a model for monitoring the catalyst, enabling better planning and maintenance or replacement of the catalytic system, thus reducing the emission of toxic gases and thus improving the quality of life, reducing the costs of treating respiratory problems caused by pollution.

Considering that the NO$_x$ sensor can be used at low or high temperatures, and is widely used in the automotive industry due to its small size, fast response, low price and long service life, the monitoring system proposed in this paper can be easily produced because it uses sensors that are already commercially available and used in the automotive industry [1,38].

**Author Contributions:** J.F., D.I.A., M.E.K.F. and L.N.B.d.A. carried out the methodology and investigation; L.M.S.C., M.E.K.F., G.G.L. and A.M.T. conceived of the project and shared in the writing, reviewing and editing. All authors have read and agreed to the published version of the manuscript.

**Funding:** This research received no external funding.

**Institutional Review Board Statement:** Not applicable.

**Informed Consent Statement:** Not applicable.

**Data Availability Statement:** Not applicable.

**Acknowledgments:** The authors thank the Capes, Fundação Araucária, and CNPq agency. The seventh author thanks CNPq for the financial support (Process: 310562/2021-0).

**Conflicts of Interest:** The authors declare no conflict of interest.

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
