# Peer review of "Catalytic Systems in the Reduction of Nitrogen Oxide Emissions in Diesel-Powered Trucks"

_sustainability, doi:10.3390/su14116662_

Round 1
Reviewer 1 Report
A revision is need before publication. The main drawbacks are listed below.
- It is not clear what “carbon (CO2)”, “ASC”, “DOC”, “COD”, “RTD”, “cDPF”, “EGTS”, “PT sensor 100” are.
- Abbreviations PM, CO2, VOC, NOx and SCR enough to determine once.
- “According to [29]” should be replace by “According to Benvenutti et al. [29]”.
- The authors write “we can estimate the emission of 256,434,158.4 tons of CO2”. I believe that the methods used by the authors do not provide such accuracy.
Author Response
We would like to thank the reviewers for spending their time in reading, reviewing, and commenting on our manuscript. Those comments are all valuable and very helpful for revising and improving our manuscript to a better scientific level.
We have studied the raised comments carefully and made corrections, which we hope that meet your requirements. Please, consider the reviewers' comments in black, the authors' answers in blue and the changes made to the paper in red.

Reviewer 2 Report
This manuscript provides an overview of the emission of toxic gases by heavy transport powered by diesel oil and the influence of the use of automotive catalysts in reducing the emission of toxic gases. However, there are some issues should be addressed before its publication.
- Does this manuscript take into account the reduction in lifetime due to damage to sensor when detecting NOXand NH3?
- Some related important works in the SCR fields were neglected in the manuscript, which should be cited, e.g. DOI: 1021/ACSCATAL.0C02567, DOI: 10.1016/J.FUEL.2012.09.046
- Whether the sensor considers the problem of gas trapping will reduce the life of the sensor?
- Whether the sensor has the ability to withstand high temperature for a long time?
- What are the advantages of the sensor presented in this manuscript over other sensors?
- Did the author consider the cost of the sensor, if the cost is too high, the scheme may not be widely used?
- The sensor mentioned in this manuscript is very practical, has such a device been made at present?If so, could it be easily produced?
Author Response

(The authors gave the same response as above.)

Reviewer 3 Report
This manuscript reported an overview of the emission of toxic gases by heavy transport powered by diesel oil and the influence of the use of automotive catalysts in reducing the emission of toxic gases. Furthermore, the application for monitoring the useful life of automotive catalysts is presented through an electronic sensing system, which makes it possible to determine the catalyst. Overall The authors report an interesting approach and the presentation of the work is clear. The objective and justification of the work are clear, and the experimental work is significant. The study is accurate and adequate.
Author Response
We would like to thank the reviewers for spending their time in reading, reviewing, and commenting on our manuscript.

Round 2
Reviewer 2 Report
The authors have revised the manuscript following all my previous suggestions, I agree for its publication in the present form